# Eglcr: Edge Structure Guidance and Scale Adaptive Attention for Iterative Stereo Matching

## ABSTRACT

Stereo matching is a pivotal technique for depth estimation and has been popularly applied in various computer vision tasks. Although many related methods have been reported recently, they still face some challenges such as significant disparity variations at object boundaries, difficult prediction at large disparity regions, and suboptimal generalization when label distribution varies between source and target domains. Therefore, we propose a stereo-matching model (i.e., EGLCR-Stereo) that utilizes edge structure information and multi-scale matching similarity features for better disparity estimation. First, we use a lightweight network to predict the initial disparity. Then, we develop a multi-scale similarity feature extraction module, incorporating adaptive attention mechanisms, to capture the fusion similarity information of stereo images across various scales. Meanwhile, we introduce an edge structure-aware module that features an iteratively optimized disparity map and a scale attention factor, aimed at accurately delineating edge information in complex scenes. After that, we employ an iterative strategy for disparity estimation, guided by the fusion similarity features across multiple scales and the detailed edge structure information. We conduct abundant experiments on some popular stereo matching datasets including Middlebury, KITTI, ETH3D, and Scene Flow. The results show that our proposed EGLCR-Stereo achieves state-of-the-art performance both in accuracy and generalization.

## KEYWORDS

Stereo matching, Edge estimation, Depth estimation, Feature extraction, Attention mechanism

## 1 INTRODUCTION

3D scene information plays a crucial role in environmental perception, and estimation of 3D scene information attracts extensive attention in many computer vision fields, such as robotic navigation, autonomous driving, and intelligent industrial monitoring [7, 11, 19], and so on. Actually, the estimation of 3D scene information always uses computer vision to calculate depth information from single, double or multi-view images. The estimation method can be mainly divided into monocular depth estimation, stereo matching, and multi-view depth estimation. Among these methods, stereo matching method is the most popularly used and researched one by the scholars and engineers, because it has simple acquisition system structure and high computational efficiency [1, 6, 8, 20].

Stereo matching method can be broadly categorized into traditional method and deep learning method. The traditional method requires to manually design the calculation strategy and the deep learning method uses the deep neural network to estimate disparity in an end-to-end manner [10, 12, 13, 41]. Compared to the traditional method, the deep learning method has better computational efficiency, robust generalization capabilities and enhanced performance. Therefore, it has gradually dominated the field of stereo

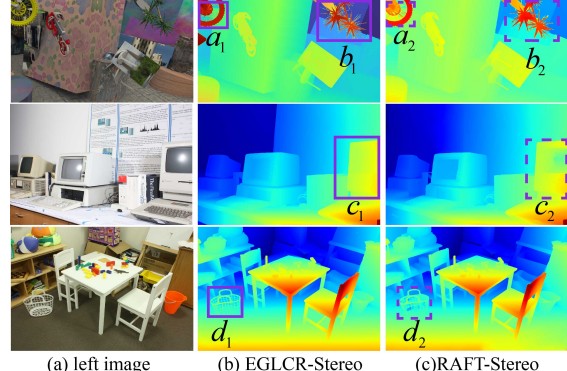

(a) left image    (b) EGLCR-Stereo    (c)RAFT-Stereo

**Figure 1: Visualization results of the estimated disparity by our proposed EGLCR-Stereo and RAFT-Stereo. Our model predicts more accurately at the edges of fine structures (see $a_1/a_2$ and $b_1/b_2$). In the areas with large disparity, our method achieves smoother and more accurate disparity (see $c_1$ and $c_2$). For cross-domain prediction, our model also outperforms RAFT-Stereo at the structural edges of objects (see $d_1$ and $d_2$).**

matching. One famous deep learning model for stereo matching is PSMNet [1], which is the representative of volume-based stereo matching models. This kind of model employs matching feature maps to construct a cost volume, and then regularizes the cost volume by some strategies to obtain the disparity map. As it uses the stacked hourglass 3D CNN to extend the regional support of context information in cost volume, it shows good performance on stereo matching. However, the volume-based method has substantial memory consumption and it may lead to a suboptimal detail prediction during cost volume regularization. To overcome this problem, another famous model for stereo matching, RAFT-Stereo [16], is proposed. It iteratively refines the disparity predictions by the contextual information to predict disparity residuals based on current disparity results, and thus obtain more detailed and precise disparity results. This kind of model using iterative refinement strategy is always called as the iterative-based stereo matching method.

Recently, an increasing number of iterative-based methods have been proposed and have achieved state-of-the-art (SOTA) performance on stereo matching. However, it still faces some challenges such as significant disparity variations at object boundaries, difficult prediction at large disparity regions, and decreased generalization when domains gap exists between the source and target domains. For example in Fig. 1(c), there are some details missing at the edges of wheel spokes (a2) and plant's periphery (b2); some inaccurate and uneven disparities appear in the large disparity area and weak texture area on the computer screen (c2); average values at the basket edges (d2) is tend to be predicted without enough details during cross-domain prediction. There are some possible reasons for these

$w_l$ : scale attention of small-scale factor

$F_{l(r)}$: matching feature map of left(right) image

**ESA:** Edge structure-aware module

**MSFEAA:** Multi-scale Similarity Feature Extraction Module Based on Adaptive Attention Mechanism

$f_d^{(k)}$: disparity feature encoded by $d^{(k)}$

$(k)$: $k^{th}$ iteration

$\Delta d^{(k)}$: residual map of disparity

$h^{(k)}$: hidden state of GRU

$e^{(k)}$: edge map

$d^{(k)}$: disparity map

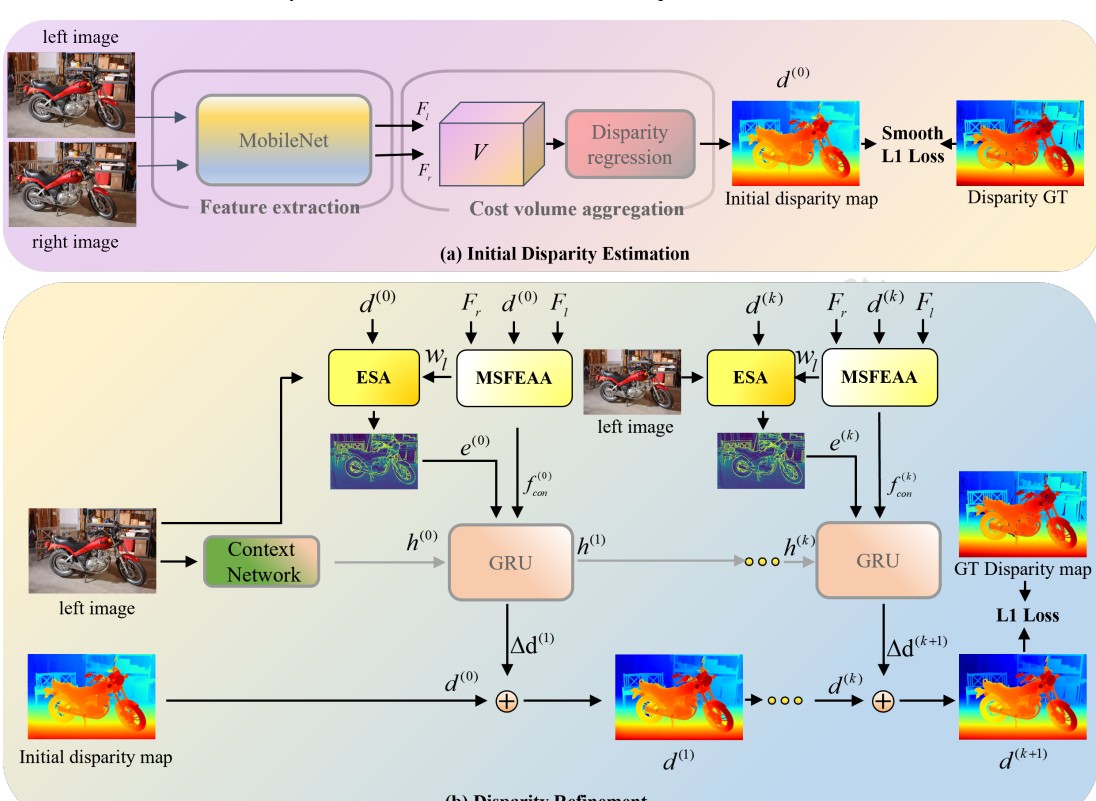

Figure 2: The pipeline of the proposed EGLCR-Stereo. The model includes two stages: initial disparity estimation and disparity refinement stage. In the first stage, we use a feature extraction module to get the matching feature of left and right view, and then we construct the cost volume and regulate the cost volume for an initial disparity map. In the second stage, we propose a multi-scale similarity feature extraction module to capture the fusion similarity information of stereo images across various scales; then, we design an edge structure-aware module with iteratively optimized dsiparity map and scale attention factor to extract refined edge structures; finally, we use an GRU unit to refine disparity through iterative-based strategy by integrating the fused multi-scale matching similarity features with the refined edge structure information.

unsatisfactory results. (1) The first reason is that during the iterative optimization process, the matching similarity features at larger scales are not effectively integrated with those at smaller scales. This often causes models to become trapped in local optima, leading to suboptimal predictions in areas with pronounced disparities. (2) The second issue is the inefficient perception of edge structure information throughout the iterative process. This inefficiency can lead to erroneous estimations and the omission of critical details along the edges of complex structures.

Therefore, in this study, we present a stereo matching model designated as EGLCR-Stereo, which leverages multi-scale matching similarity features alongside refined edge structure information for disparity estimation. The proposed model adeptly addresses the aforementioned challenges and consistently delivers satisfactory outcomes. (Please see Fig. 1(b) for example). The main contributions are as follows.

(1) We develop a multi-scale similarity feature extraction module designed to capture the fusion similarity information of stereo images across various scales. This module encompasses a small-scale matching similarity feature extraction component employing group correlation, and a large-scale counterpart utilizing inner product operations to generate multi-scale similarity features. Notably, we introduce a scale-adaptive attention mechanism that dynamically adjusts the weights of large and small-scale matching similarity features based on the current matching state. This innovation significantly enhances the convergence speed of the iterative disparity refinement process.

(2) We introduce an edge structure-aware module that incorporates an iteratively optimized disparity map and a scale attention factor. In this module, we design a disparity head that processes the iteratively optimized disparity and a texture head that handles texture information. Additionally, we use the scale attention factor

from the MSFEAA module to fuse these elements, and then design an edge head for refined edge structures.

(3) We employ a recurrent neural network to continuously predict residual disparities by integrating multi-scale matching similarity features with refined edge structures. Our model surpasses the baseline model on SceneFlow, Middlebury and ETH3D datasets, achieving state-of-the-art (SOTA) performance both on accuracy and cross-domain generalization. Our code will be open-sourced on Github.

## 2 RELATED WORKS

Recently, deep learning methods have been dominated the field of stereo matching and achieved impressive performance on most of standard benchmarks. These methods are mostly improved from the perspectives of cost volume aggregation, iterative disparity map optimization, and feature-guided stereo matching methods.

**Volume-aggregation based Method.** To enhance the information fusion from left and right views and further improve the performance of stereo matching, many researchers [1, 10, 15, 31, 32, 35, 40] consider the improvements from feature extraction and cost volume aggregation. These methods always include four steps: feature extraction, cost volume construction, cost volume aggregation, and disparity map calculation process. For example, Liu et al. [17] employed a local similarity pattern to enhance the matching feature before the cost volume construction, and they reported that their model could effectively extract local structural information and improve the matching performance in the regions with edge structures. In the matching cost volume construction, Xu et al. [33] employed a fusion cost volume derived from the amalgamation of the group-wise cost volume and the concatenated cost volume to optimally assimilate the global and local information, and then regularize this cost volume to obtain disparity map. Although the above volume-aggregation based methods improve the matching performance, they still incur large calculations of cost volume regularization and the prediction errors in some complex areas.

**Iterative-based Method.** To overcome the above limitations of volume-aggregation based method, some researchers [16, 29] introduced the iterative optimization strategy for stereo matching. This kind of method gradually generates disparity maps from coarse to fine. For example, RAFT-Stereo [16] employed a Gated Recurrent Unit (GRU) that is embedded in Convolutional Neural Network (CNN) for disparity map refinement, and improved the disparity prediction and inference speed. CREStereo [14] took the iterative optimization strategy and introduced an adaptive group correlation layer to alleviate the impact of imperfect rectification. They showed that their proposed model effectively improved depth estimation performance in cases where left and right view corrections are imperfect. In order to improve the model's understanding of contextual information, IGEV-Stereo [34] utilized the principles of residual learning and iterative enhancement and proposed a geometric encoding volume to continuously optimize the disparity map. By adopting an iterative optimization strategy, the iterative-based method can obtain more precise disparity prediction results than the volume-aggregation based method.

**Feature-guided based method.** Beyond the methods previously discussed, several techniques [2, 9, 39] supervise disparity

generation using supplementary feature information. Edge Stereo [27] integrated a dedicated edge prediction branch for predicting object edges and disparity related to the left view, and guided edge graph generation with edge-aware perception smoothness loss. Dai et al. [4] used the color and gradient consistency between the corresponding pixels in the view as supervisory signals to guide the generation of disparity maps. Xu et al. [37] used semantic information from corresponding pixels as supervisory signals to establish a semantic consistency loss function for model training, and enhanced the robustness of the model. It is crucial to use additional information to supervise the disparity generation in specific domains, but model performance depends on the effectiveness of the information.

In this article, we propose an EGLCR-Stereo model using residual learning and multitasking learning principles. It effectively utilizes the multi-scale similarity and edge structure information to iteratively refines the disparity predictions from coarse to fine.

## 3 EGLCR-STEREO

Fig. 2 shows the network structure of the proposed EGLCR-Stereo model. It includes initial disparity estimation and disparity refinement stage, where the disparity refinement is our major contribution.

### 3.1 Initial Disparity Estimation

Giving the left and right input images $I_l, I_r \in \mathbb{R}^{3 \times H \times W}$, we adopt a MobileNet [22] pre-trained on ImageNet [5] to extract the feature maps $f_l, f_r \in \mathbb{R}^{C \times \frac{H}{4} \times \frac{W}{4}}$. Then, we use a channel fusion module with a convolution kernel of 1 to enhance $f_l, f_r$, and then obtain the left and right enhanced matching features $F_l$ and $F_r$. After that, we use $F_l$ and $F_r$ to construct a group-wise correlation cost volume $V$ and regularize it by a lightweight 3D convolution module to get the probability volume $\overline{V} \in \mathbb{R}^{d \times \frac{h}{4} \times \frac{w}{4}}$.

The initial disparity map is calculated by the expectation of the probability volume along the disparity dimension:

$$d^{(0)} = \sum_{d=0}^{d_{max}} d * \overline{V}(d) \ , \tag{1}$$

where $d^{(0)} \in \mathbb{R}^{\frac{h}{4} \times \frac{w}{4}}$.

### 3.2 Multi-scale Similarity Feature Extraction Module Based on Adaptive Attention Mechanism

We design a multi-scale similarity feature extraction module with an adaptive attention mechanism, including a large-scale matching similarity feature extraction module, a small-scale matching similarity feature extraction module, and a scale adaptive attention module, as shown in Fig. 4 (a).

**Small-scale matching similarity feature extraction module.** To refine the disparity result, we search the optimal disparity within a local neighbourhood for each candidate match point. We use a small-scale matching similarity feature extraction module (shown in Fig. 3(b)) to extract the features of the neighbouring pixels. For the standard pixel coordinates $(i, j)$ of the left image and the initial disparity $d^{(0)}(i, j)$, the coordinate position of the matching point

in the right image can be determined as

$$o_r = (i, j - d^{(0)}(i, j)) \quad . \tag{2}$$

After that, we sample the corresponding feature of $F_r$ along the disparity direction within the range $\delta$. Then, we concatenate the reference feature $F_l(i, j)$ with the feature sampled from the right view to obtain the local similarity feature $f_{loc}(i, j)$. The above process can be expressed by the formula:

$$f_{sma}(i, j) = \mathbf{Corr}_g\{F_l(i, j), F_r(i, j_n) | j_n \text{ in } \mathbf{N}_l\} \quad , \tag{3}$$

where $\mathbf{N}_l = \{j - d^{(k)}(i, j) - \delta, ..., j - d^{(k)}(i, j) + \delta\}$, and $\mathbf{Corr}_g$ represents group-wise correlation calculation. Computing the similarity between matching features in high-dimensional space in the form of group correlation can characterize the similarity of matching feature vectors at different groups (e.g., color, texture, etc.) and improve the robustness of matching results, which can provide more information for subsequent disparity refinement.

**Large-scale matching similarity feature extraction module.** It is insufficient to utilize only small-scale matching similarity feature for disparity refinement. This is because that if a substantial deviation occurs between the disparity in the current stage with the ground truth, the local contextual information will restrict the scope to obtain the global optimization. Therefore, we apply the large-scale perception module for a large receptive field, and help the model escape from the local solution and obtain the global optimal solution. Following by Raft-Stereo, we select the all-pair similarity measure as the global perception feature to guide the iterative refinement process. As shown in Fig. 3 (a), given the left and right feature maps $F_{l(r)}$ and disparity map $d^{(k)}$, we sample the similarity along the initial disparity direction within the neighborhoods to obtain the global feature map:

$$f_{lar}(i, j) = \mathbf{Inner}\{\langle F_l(i, j), F_r(i, j_n) \rangle | j_n \text{ in } \mathbf{N}_g\} \quad , \tag{4}$$

where $f_{glo} \in \mathbb{R}^{\mathbf{N}_g \times \frac{h}{4} \times \frac{w}{4}}$, $\mathbf{N}_g = \{j - d^{(k)}(i, j) - 8\delta, ..., j - d^{(k)}(i, j) + 8\delta\}$ indicates that the large-scale matching similarity feature extraction module considers a wider neighborhood information than that in the small-scale sampling module, $\mathbf{Inner}\langle \cdot, \cdot \rangle$ denotes the inner product operation that not only enhances computational efficiency but also optimizes memory utilization, compared to the group correlation operation. To further expand the reception field, we downsample the similarity matrices of the left and right feature maps along the disparity channel to construct a pyramid of similarity feature maps, and then neighborhood sampling operations are performed at each scale to obtain large-scale matching similarity features, as shown in Fig. 3 (a).

**Scale adaptive attention module.** When the predicted disparity is close to the global optimal value, small-scale matching similarity feature is more important than the large-scale matching similarity features. Conversely, when the predicted disparity is far from the global optimal value, the large-scale features should be more carefully treated. Therefore, we introduce a scale adaptive attention module to integrate the large and small scale matching information. As shown in Fig. 4 (b), We calculate a scale attention i.e.,

$$W_s = \mathrm{softmax}(\mathcal{F}_{att}(f_{lar})), \tag{5}$$

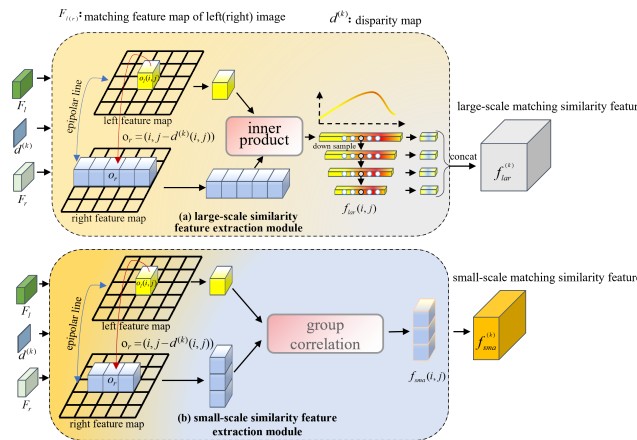

Figure 3: Large and small-scale matching similarity feature extraction modules.

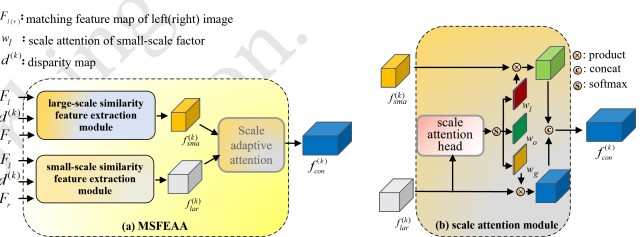

Figure 4: Multi-scale similarity feature extraction module based on adaptive attention mechanism.

where $W_s \in \mathbb{R}^{3 \times \frac{h}{4} \times \frac{w}{4}} = [w_l, w_g, w_o]$. Note that $w_l$, $w_g$, and $w_o$ represents the weights of small-scale information, large-scale information, and the likelihood of a point being an outlier, respectively. After that, we weight the small and large scale matching similarity features by $w_l$ and $w_g$, and then concatenate them along the feature channel to obtain the multi-scale fusing context feature $f_{con}$ as:

$$f_{con} = \mathrm{Concat}\{w_l * f_{loc}, w_g * f_{glo}\} \tag{6}$$

## 3.3 Edge Structure-aware module with iteratively optimized disparity and scale attention factor

In this module, we use both texture and disparity feature to estimate the edge map of the left view as shown in Fig. 5. Given the substantial semantic information contained in the left feature map, we utilize $f_l$ by a texture head to extract the texture feature of the left image. Then, we use a disparity head to extract disparity features on the disparity map at each iteration. After that, we use a edge head to calculate the edge map by integrating the texture and disparity feature. The above processes can be indicated as:

$$\begin{cases} w_t = (1 - \sigma) * (1 - w_l) \\ w_d = \sigma * w_l \\ e^{(k)} = \mathcal{F}_e(w_t * f_{tex}, w_d * f_{disp})), \end{cases} \tag{7}$$

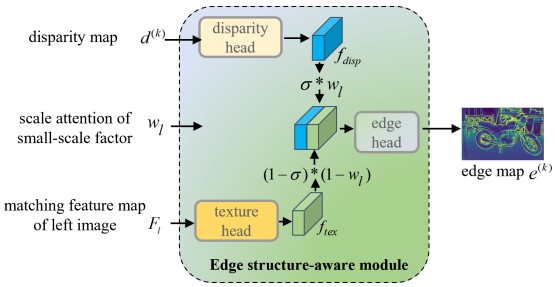

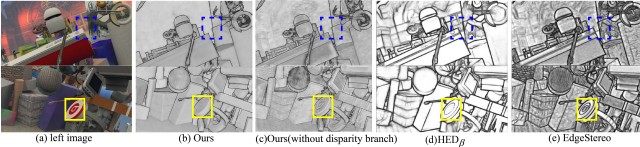

(a) left image    (b) Ours    (c)Ours(without disparity branch)    (d)$HED_\beta$    (e) EdgeStereo

Figure 5: Edge structure-aware module with iteratively optimized disparity and scale attention factor.

where $\mathcal{F}_e$ includes several convolutional layers and each convolutional layer is accompanied by a normalization and activation layer; Meanwhile, we implement an adaptive weighting strategy to maximize the efficacy of the disparity and texture branch. We introduce a parameter $\sigma$ to incrementally amplify the impact of disparity branch throughout the iterative procedure as $\sigma = \frac{k}{K+1}$, where $k$ and $K$ denote the current iteration number and total iteration number, respectively. As the number of iterations increases, the weight of the disparity branch gradually increases to reduce the impact of inaccurate disparity maps during early iterations.

### 3.4 GRU-based disparity iterative optimization.

The iterative optimization considers the disparity and edge structure feature from the previous iteration, as well as multi-scale matching similarity information. To extract high frequency features from disparity and edge map as:

$$f_d^{(k)}, f_e^{(k)} = \mathcal{F}_h(d^{(k)}, e^{(k)}) \qquad (8)$$

where $f_d^{(k)} \in \mathbb{R}^{c_d \times \frac{h}{4} \times \frac{w}{4}}$ and $f_e^{(k)} \in \mathbb{R}^{c_e \times \frac{h}{4} \times \frac{w}{4}}$.

After that, we combine high frequency features, matching similarity feature to update the hidden layer state of GRU and calculate the residual value of the disparity $\triangle d$ at each iteration. The process at the $k^{th}$ iteration can be indicated as:

$$\begin{cases} h^{(k+1)}, \triangle d^{(k+1)} = \mathbf{GRU}(h^{(k)}, f_d^{(k)}, f_e^{(k)}, f_{con}^{(k)}) \\ d^{(k+1)} = d^{(k)} + \triangle d^{(k+1)} \end{cases} \qquad (9)$$

**Loss function.** The loss function includes edge loss $\mathcal{L}_e$ and disparity loss $\mathcal{L}_d$. Considering the quantity imbalance between the pixel points in edge and non-edge regions, we use the weighted binary cross-entropy loss to represent the edge loss:

$$\mathcal{L}_e = \sum_{k=0}^{K-1} \gamma^{K-1-k} \mathbf{BCE}(e^{(k)}, e_{gt}, w), \qquad (10)$$

where $e^{(k)}$ is the edge map predicted at the $k^{th}$ iteration, $w$ is the ratio of the number of edge pixels to non-edge pixels in the image, $e_{gt}$ is the ground truth of edge map and $\gamma^{K-1-k}$ is the weight at each iteration. Besides, we use $L_1$ loss and smooth $L_1$ loss to represent the disparity loss $\mathcal{L}_d$ as:

$$\mathcal{L}_d = \mathbf{Smooth\ L1}(d^{(0)}, d_{gt}) + \sum_{k=1}^{K-1} \gamma^{K-1-k}|d^{(k)} - d_{gt}|, \qquad (11)$$

Figure 6: The first column denotes the original RGB images, the second column denotes the edge maps predicted by our adaptive weighted edge estimation module, the third column denotes the edge maps without disparity encoding features, the fourth column denotes the edge maps predicted by the $HED_\beta$ method, and the fifth column denotes the edge maps predicted by EdgeStereo. Our model obtains more accurate boundaries of the real object than other models.

Therefore, the total loss can be indicated as:

$$\mathcal{L} = \mathcal{L}_d + \lambda\mathcal{L}_e, \qquad (12)$$

where $\lambda$ is a weight factor to balance the edge loss and disparity loss.

## 4 EXPERIMENT

### 4.1 Dataset and Evaluation Metrics

**Scene Flow** [20] is a synthetic stereo matching benchmark with ground truth. The dataset has 35,454 training images and 4,370 test images, with a image resolution of $960 \times 540$ pixels.

**KITTI15** [21] consists of real-world stereo images captured by a vehicle-mounted stereo camera. The dataset provides sparse ground truth disparity maps. The KITTI15 dataset contains 200 training and test images, and the images have a resolution of $375 \times 1242$ pixels.

**Middlebury2014** [23] is a high-quality benchmark for stereo matching algorithms. It includes 15 training and 15 test samples. Each sample provides left and right view images at full (F), half (H), and quarter (Q) resolutions, as well as dense ground truth disparity maps.

**ETH3D** is a benchmark for stereo matching algorithms. It includes 27 training and 20 test grayscale image samples. The dataset provides a platform for assessing the generalization capabilities of different stereo matching models.

### 4.2 Implementation Details

The proposed EGLCR-Stereo is implemented by the PyTorch framework and trained on a server equipped with multiple 3090 graphics cards. During the first stage of training, we employ the AdamW optimizer [18] with an initial learning rate of 2e-4 and use the One-Cycle learning rate update strategy. The model is initially trained for 200k steps on the SceneFlow dataset. Then, the weight parameters related to edge prediction are frozen and only the parameters for disparity prediction are fine-tuned for an additional 100k steps. The input image size is randomly cropped to $320 \times 736$. On the Middlebury datasets, we use an initial learning rate of 1e-5, and freeze the weight parameters related to edge prediction and fine-tune the model for 10K steps with only disparity loss. The number of iterations for GRU disparity refinement is set to 20 and 32 during

**Table 1: Ablation study of the EGLCR-stereo on Sceneflow test dataset. ESA, SSMS and LSMS denote the edge structure-aware module, small and large scale matching similarity feature extraction module respectively.**

| | Model | ESA | SSMS | LSMS | diaparity<192 | | | | all disparity | | | |
|---|---|---|---|---|---|---|---|---|---|---|---|---|
| | | | | | >1px (%) | >3px (%) | >5px (%) | EPE (px) | >1px (%) | >3px (%) | >5px (%) | EPE (px) |
| (a) | RAFT [16] | ✗ | ✗ | ✗ | 7.29 | 3.41 | 2.36 | 0.72 | 7.74 | 3.73 | 2.62 | 0.81 |
| (b) | RAFT+AE | ✓ | ✗ | ✗ | 6.07 | 2.84 | 1.96 | 0.57 | 6.74 | 3.24 | 2.37 | 0.80 |
| (c) | RAFT+AE+LCS | ✓ | ✓ | ✗ | 5.71 | 2.63 | 1.85 | 0.53 | 6.36 | 3.09 | 2.13 | 0.78 |
| (d) | **RAFT + AE + LCS + GP EGLCR-Stereo(ours)** | ✓ | ✓ | ✓ | **5.19** | **2.52** | **1.76** | **0.50** | **5.60** | **2.82** | **2.01** | **0.76** |

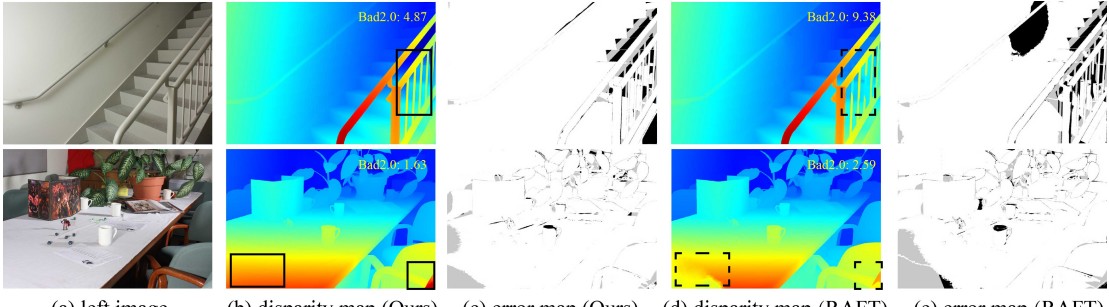

(a) left image    (b) disparity map (Ours)    (c) error map (Ours)    (d) disparity map (RAFT)    (e) error map (RAFT)

**Figure 7: Results on Middlebury online leaderboard. The first column denotes the original RGB images, the second column denotes the disparity maps predicted by our EGLCR-Stereo, the third column denotes the bad2.0 error maps between our results and the ground truth. The fourth and fifth columns denotes the disparity maps and error maps of RAFT-Stereo, respectively. Our predictions demonstrate superior performance to the benchmarked models in textureless regions and areas with significant depth variations.**

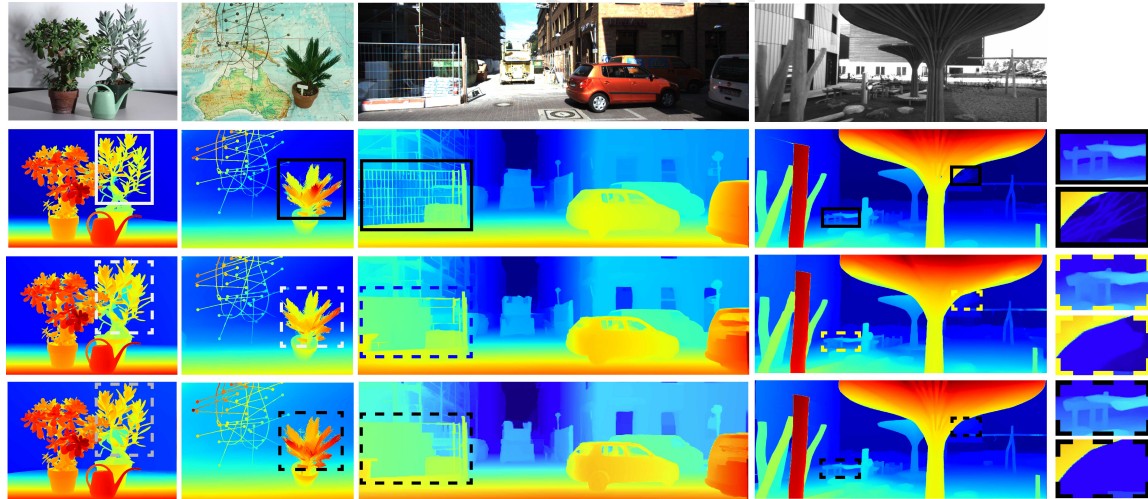

**Figure 8: Generalization results on Middlebury 2014, Kitti2015 and ETH3D. The second, third and fourth row show the results of our EGLCR-Stereo, RAFT and IGEV-Stereo, respectively. The last column shows the detailed results in the fourth column. Compared to other models, our EGLCR-Stereo shows excellent ability to retain complex details at the edges of objects.**

**Table 2: Quantitative results on Sceneflow test dataset.**

| Model | diaparity<192 | | all disparity | |
|---|---|---|---|---|
| | D1 (%) | EPE (px) | D1 (%) | EPE (px) |
| PSMNet [1] | 3.09 | 0.92 | 3.60 | 1.03 |
| RAFT [16] | 3.41 | 0.72 | 3.73 | 0.81 |
| **EGLCR-Stereo (ours)** | **2.52** | **0.50** | **2.82** | **0.76** |

the model training and testing, respectively. In all experiments, RGB values of input images are normalized to lie between -1 and 1.

## 4.3 Ablation Studies

We conducted the ablation studies on the edge structure-aware module, small-scale matching similarity feature extraction module and large-scale matching similarity feature extraction module.

**Edge structure-aware module.** To demonstrate the effectiveness of the edge structure-aware module, we firstly compared the RAFT models with/without edge structure-aware module for disparity estimation. The results are shown in Tab. 1 (a) and (b). The

**Table 3: Quantitative results on Middlebury leaderboard benchmark. Bold indicates the best metric, and underline indicates the second best metric.**

| Method | Bad0.5 | Bad1.0 | AvgErr | RMS | A50 | A99 |
|---|---|---|---|---|---|---|
| EdgeStereo [26] | 55.6 | 32.4 | 2.68 | 9.84 | 0.72 | 40.8 |
| AANet++ [35] | 48.6 | 25.5 | 6.37 | 23.5 | 0.56 | 114 |
| CFNet [24] | 43.7 | 19.6 | 3.49 | 15.4 | 0.48 | 77.6 |
| AdaStereo [25] | 55.0 | 29.5 | 2.22 | 10.2 | 0.65 | 40.6 |
| HSMNet [38] | 50.7 | 24.6 | 2.07 | 10.3 | 0.56 | 39.2 |
| HITNet [28] | 34.2 | 13.3 | 1.71 | 9.97 | 0.40 | 30.2 |
| LEAStereo [3] | 49.5 | 20.8 | 1.43 | 8.11 | 0.53 | 20.2 |
| CroCo v2 [30] | 40.6 | 16.9 | 1.76 | 8.91 | 0.39 | 35.0 |
| GMStereo [36] | 51.5 | 23.6 | 1.31 | **6.45** | 0.58 | 16.8 |
| CREStereo [14] | 28.0 | **8.25** | 1.15 | 7.70 | 0.28 | 22.9 |
| IGEV-Stereo [34] | 32.4 | 9.41 | 2.89 | 12.8 | 0.32 | 43.0 |
| RAFT-Stereo [16] | 27.7 | 9.37 | 1.27 | 8.41 | **0.26** | 21.7 |
| **EGLCR-Stereo (ours)** | **27.6** | 8.59 | **1.05** | 7.04 | 0.27 | **15.0** |

**Table 4: Quantitative results on ETH3D leaderboard benchmark. Bold indicates the best metric, and underline indicates the second best metric.**

| Method | Bad0.5 | Bad1.0 | AvgErr | A90 | A95 |
|---|---|---|---|---|---|
| EdgeStereo [26] | 18.75 | 6.76 | 0.39 | 0.76 | 1.11 |
| AANet++ [35] | 13.16 | 5.01 | 0.31 | 0.57 | 1.16 |
| CFNet [24] | 10.67 | 3.72 | 0.27 | 0.51 | 0.96 |
| AdaStereo [25] | 10.22 | 3.09 | 0.24 | 0.50 | 0.70 |
| RAFT-Stereo [16] | 7.04 | 2.44 | 0.18 | 0.39 | 0.57 |
| DIP-Stereo [42] | 6.74 | 1.97 | 0.18 | 0.39 | 0.62 |
| GMStereo [36] | 5.94 | 1.83 | 0.19 | 0.39 | 0.55 |
| CroCo v2 [30] | 3.27 | 0.99 | 0.14 | 0.26 | 0.40 |
| IGEV-Stereo [34] | 3.52 | 1.12 | 0.14 | 0.29 | 0.43 |
| CREStereo [14] | 3.58 | **0.98** | **0.13** | 0.28 | 0.39 |
| **EGLCR-Stereo (ours)** | **2.92** | 1.04 | **0.13** | **0.25** | **0.37** |

1-pixel error rate (disparity<192) decreases from 7.29 to 6.07 by utilizing the edge structure-aware module, which shows that the edge structure-aware module can increase the performance of disparity estimation. Besides, as we import the disparity information in our edge structure-aware module that is a main difference to other edge estimation modules (i.e., $HED_\beta$ and EdgeStereo),we compared the EGLCR with/without disparity branch, as well as $HED_\beta$ and EdgeStereo. The results are shown in Fig. 6, and indicate that our model obtains more accurate boundaries of the real object than other models.

**Small-scale matching similarity feature extraction module.** We compared the RAFT models with/without the small-scale matching similarity feature extraction module for disparity estimation. The results are shown in Tab. 1 (b) and (c). The integration of small-scale matching similarity feature leads to a reduction in 1-pixel error rate (disparity<192) metric from 6.07 to 5.71 and End Point Error (EPE:192) metric decreased by 7%, that is because the small-scale matching similarity feature could provide more detailed information for disparity refinement.

**Large-scale matching similarity feature extraction module.** We compared the RAFT models with/without the large-scale matching similarity feature extraction module for disparity estimation. The results are shown in Tab. 1 (c) and (d). The integration of large-scale matching similarity feature leads to a reduction in the 1-pixel error rate (all disparity) metric by 11.9%, that is because the large-scale matching similarity feature can enhance the global perception ability of model and yield more resilient predictive outcomes in the areas with large disparity.

**Table 5: Quantitative results on KITTI leaderboard. (Red: the best, Blue: the second, Green: the third, ref denotes reflective region).**

| Method | KITTI15 | | | KITTI12 | | | | | |
|---|---|---|---|---|---|---|---|---|---|
| | D1-all | D1-fg | D1-bg | 3-noc | 4-noc | 5-noc | 3-noc (ref) | 4-noc (ref) | 5-noc (ref) |
| HITNet | 1.98 | 3.20 | 1.74 | 1.41 | 1.14 | 0.96 | 5.91 | 4.04 | 2.95 |
| RAFT-Stereo | 1.96 | 2.89 | 1.75 | 1.30 | 1.03 | 0.86 | 5.40 | 4.24 | 3.60 |
| CFNet | 1.88 | 3.56 | 1.54 | 1.23 | 0.92 | 0.74 | 5.96 | 4.24 | 3.28 |
| LaC+GANet | 1.67 | 2.83 | 1.44 | 1.05 | 0.80 | 0.65 | 6.02 | 4.15 | 3.22 |
| PCWNet | 1.67 | 3.16 | 1.37 | 1.04 | 0.78 | 0.63 | 4.99 | 3.38 | 2.54 |
| ACVNet | 1.65 | 3.07 | 1.37 | 1.13 | 0.86 | 0.71 | 7.03 | 5.18 | 4.14 |
| DLNR | 1.76 | 2.59 | 1.60 | N/A | N/A | N/A | N/A | N/A | N/A |
| IGEV-Stereo | 1.59 | 2.67 | 1.38 | 1.12 | 0.88 | 0.73 | 4.35 | 3.16 | 2.55 |
| EGLCR-Stereo(ours) | 1.60 | 2.71 | 1.38 | 1.09 | 0.81 | 0.65 | 3.62 | 2.51 | 1.92 |

## 4.4 Comparisons with the State-of-the-art

We compare EGLCR-Stereo with several SOTA methods on the SceneFlow, Middlebury and ETH3D datasets. For SceneFlow, our proposed model demonstrates state-of-the-art (SOTA) performance in the prediction of disparities less than 192, achieving an EPE of 0.50 and a D1 error rate (3px) of 2.52. When it comes to large disparity prediction, our model shows significant improvements over existing methods, outperforming RAFT-Stereo by 6% and leading PSMNet by 26.2% in the EPE metric. These results are detailed in Tab. 2. The superior performance of our EGLCR-Stereo can be attributed to the incorporation of an edge structure-aware module and multi-scale matching similarity features, which enhances the model's ability to handle detailed predictions and large textureless areas.

On the MiddleBury dataset, our method outperforms other methods on multiple metrics. At the time of writing, our EGLCR-Stereo achieves rank $1^{st}$ on the avgerr (average absolute error in pixels) and A99 (99-percent error quantile in pixels) metrics, surpassing all other existing methods on the Middlebury online dataset leaderboard, shown in Tab.3. In addition, our EGLCR-Stereo model surpasses RAFT-Stereo and IGEV-Stereo by 16.5% on the bad2.0 (percentage of bad pixels whose error is greater than 2.0) metric. Furthermore, according to the disparity map generated from the online list in Fig. 7, it is clear that our model demonstrates better performance on the large-disparity prediction and object-edge structural perception than the RAFT-Stereo.

For ETH3D and KITTI, our EGLCR-Stereo surpasses recent methodologies such as IGEV-Stereo, CREStereo and CroCo, and sets a new standard in terms of the proportion of pixels with an error exceeding 0.5px (Bad0.5) and the average mean error (AvgErr), as detailed in Tab. 4. Meanwhile, the bad0.5 metric for non-occluded pixels has been enhanced by a minimum of 10% than other methods. Table.5 shows results of quantity comparison on KITTI dataset. Our EGLCR-Stereo achieves the state-of-the-art performance among all the published models, and especially obtains remarkable improvements in the metrics of the reflective region (i.e., 3/4/5-noc(ref)).

## 4.5 Zero-Shot Generalization

It is a challenge to acquire disparity labels in the real-world, and the generalization capability is a critical metric for model evaluation. To assess this capability, we trained the EGLCR-Stereo on the SceneFlow dataset and then tested the model's performance on the MiddleBury2014 and ETH3D datasets. The results are shown in Tab. 6 and Fig. 8. On the ETH3D dataset, our approach outperforms

left image

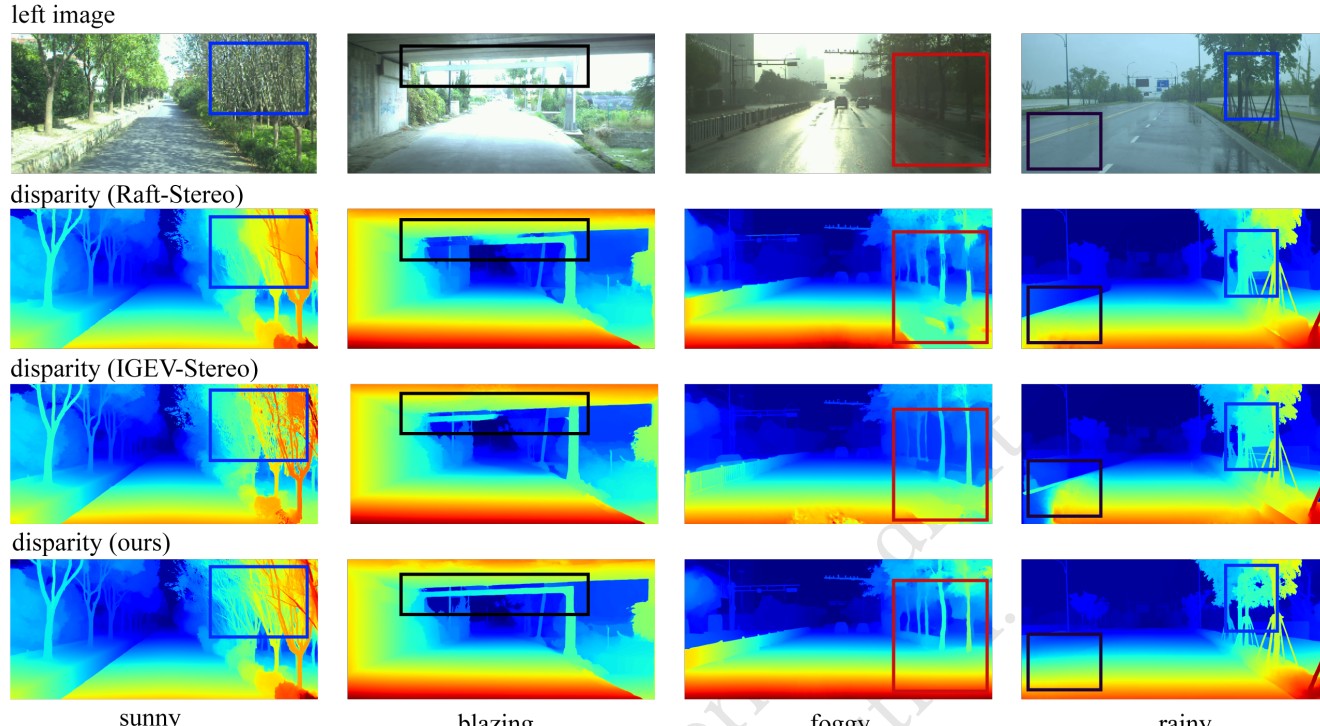

disparity (Raft-Stereo)

disparity (IGEV-Stereo)

disparity (ours)

| sunny | blazing | foggy | rainy |

**Figure 9: Disparity generalization visualization results in different real scenarios. Our method can obtain robust and accurate disparity prediction results compared to other methods in different real-world environments.**

other state-of-the-art (SOTA) methods in terms of the 1-pixel error rate metric. Besides, for the quarter-resolution images on the Middlebury dataset, our proposed model surpassed RAFT by about 4% in the 2-pixel error rate metric. Especially, when we escalate the resolution of the predicted images (the disparity range of the target domain also increases), the advantages of our model become more apparent, outstripping RAFT directly by margins of 14.6% and 12.2% for half and full resolutions, respectively. In addition, our model exhibits a robust convergence performance in disparity prediction than the baseline model. As shown in Tab. 7, the disparity prediction result at only one iteration in the inference phase exceeds the baseline model by 62.3%. This superior performance is attributed to the scale attention module and wide receptive field of our model, which helps to avoid local optimal disparity points during the iterative optimization process. Furthermore, Fig.9 displays the predicted disparities on some typical real-world scenes of StereoDriving. As shown in the rectangular box, our EGLCR-stereo obtains more satisfactory disparity details.

## 5 CONCLUSION

In this article, we introduce a novel stereo-matching model named EGLCR-Stereo, designed for disparity estimation. The model employs an iterative disparity optimization strategy. In the EGLCR-Stereo framework, we develop a multi-scale similarity feature extraction module along with an edge structure-aware module. These modules are engineered to extract similarity features from stereo images across multiple scales and to refine edge structures. The multi-scale similarity features and the refined edge structures play a

**Table 6: Generalization experiments from synthetic to real (Bold denotes best, underline denotes second best, 3px for KITTI, 2px for Middlebury, 1px for ETH3D).**

| Model | Middlebury | | | ETH3D | KITTI15 |
|---|---|---|---|---|---|
| | full | half | quarter | | |
| GwcNet | 47.1 | 34.2 | 18.1 | 12.8 | 22.7 |
| PSMNet | 39.5 | 25.1 | 14.2 | 10.2 | 16.3 |
| GANet | 32.2 | 20.3 | 11.2 | 6.5 | 11.7 |
| DSMNet | 21.8 | 13.8 | 8.1 | 6.2 | 6.50 |
| RAFT-Stereo | 12.2 | 8.9 | 7.6 | 3.2 | 5.74 |
| IGEV-Stereo | 15.1 | **7.2** | **6.2** | 3.6 | 6.04 |
| **EGLCR-Stereo (ours)** | **10.7** | 7.6 | 7.3 | **2.6** | **5.15** |

**Table 7: Iteration analysis in the model inference on the Middlebury training dataset. The 2-pixel error rate (%) is used as the experimental metric.**

| Model | Iterations | | | | | |
|---|---|---|---|---|---|---|
| | 1 | 2 | 3 | 4 | 8 | 32 |
| RAFT-Stereo [16] | 40.6 | 19.8 | 15.0 | 13.2 | 11.4 | 8.9 |
| **EGLCR (ours)** | **15.3** | **14.3** | **13.3** | **12.5** | **10.2** | **7.6** |

pivotal role as they systematically guide the disparity optimization process at each iteration. Our approach effectively navigates significant disparity variations at object boundaries, improves predictions in regions of large disparities, and enhances generalization across varying label distributions in source and target domains. Empirical results on benchmark datasets, including Middlebury, ETH3D, and Scene Flow, demonstrate that EGLCR-Stereo achieves state-of-the-art performance in terms of accuracy and generalization in disparity prediction.

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
