# OpenReview forum: "Eglcr: Edge Structure Guidance and Scale Adaptive Attention for Iterative Stereo Matching"
_acmmm.org/ACMMM/2024/Conference — MM2024 Poster_

### Official Review · Reviewer_5ybF · 2024-05-22

**Rating:** 3
**Confidence:** 3

**Summary:**

This paper proposes an iterative-based stereo matching method named EGLCR-Stereo to improve the accuracy of disparity estimation. This model mainly contributes a multi-scale similarity feature extraction module and an edge structure-aware module to the baseline method (RAFT) in the disparity refinement process. Experiments on four popular stereo matching datasets, including Middlebury, ETH3D, and Scene Flow, show the effectiveness of EGLCR-Stereo.

**Strengths:**

+ Technically, the usage of multi-scale feature and edge estimation is effective for the iterative-based disparity prediction, which is proved by the ablation study.

+ Evaluation on accuracy is adequate. As shown in the qualitative results, the EGLCR-Stereo successfully improves the disparity estimation accuracy in three kinds of scenes: the edges of fine structures, areas with large disparity, and cross-domain prediction.

**Limitations:**

- Lack of novelty. The idea of using multi-scale feature to enlarge the receptive field and capture global information in stereo matching has been proposed by "Edge supervision and multi-scale cost volume for stereo matching" (Image and Vision Computing, 2022, 117: 104336). While the difference between this paper and the proposed method is not clarified.

- Writing needs improvement.
1) The contributions are not concisely summarized. In the contribution paragraphs, there are previously proposed techniques, e.g. group correlation [6] in line 219, which makes the novelty of this paper confusing.
2) There are many unexplained terms. E.g. in Table 1, the 'SSMS' and 'LSMS' are not explained. In Figure 2, the 'context network' is not explained or mentioned in other place of the paper. In Eq. (5), the $\mathcal{F}_{att}$ is not explained.
3) There are some inaccurate expressions leading to misunderstanding. E.g. in Figure 2, the $f_{con}^{(k)}$ may be miswritten and should be $f_{d}^{(k)}$.
4) References are not adequate. E.g. the baseline model (RAFT) is not explicitly cited in the main paper, though we can read it from Table 1. And the reference of the ETH3D dataset is lacked.

- Lack of evaluation on the inference speed of the proposed method.

**Suitability:**

2

---

### Official Review · Reviewer_LgYe · 2024-05-23

**Rating:** 4
**Confidence:** 3

**Summary:**

The paper introduces the EGLCR-Stereo model for stereo matching, which can enhance disparity estimation by leveraging edge structure information and adaptive scale attention mechanisms. This model is designed to address challenges such as disparity variations at object boundaries and performance under large disparity conditions. It consists of a multi-scale similarity feature extraction module, and an edge structure-aware module, and employs GRU-based iterative optimization for refining disparity estimations. The proposed approach is validated on multiple datasets, demonstrating superior performance in terms of accuracy and generalization compared to existing methods.

**Strengths:**

1. The model is rigorously tested across multiple datasets, including both synthetic and real-world scenarios, demonstrating robustness and adaptability. The inclusion of ablation studies further strengthens the validity of the proposed enhancements.

2. EGLCR-Stereo achieves good results on several benchmarks, outperforming existing methods in terms of both accuracy and the ability to generalize across different domains. This is corroborated by quantitative results showing improvements over the baseline models.

**Limitations:**

1. The utilization of edge information to optimize stereo matching is not a novel concept, as previously explored in works like EdgeStereo. In this paper, the integration of edge information into the Raft-Stereo framework, coupled with a basic attention feature fusion mechanism, appears to be a straightforward incremental work that may show few contributions to the stereo-matching community.

2. The model appears to be computationally intensive, potentially limiting its application in real-time or on-device scenarios. However, the paper lacks a detailed discussion of the actual computational costs, such as memory requirements and execution time. Given the fact that many "so-called " new methods based on the Raft-Stereo-like framework are known to be computationally demanding, the additional components introduced in this model, such as multi-scale features, might exacerbate this issue. It would be beneficial for the authors to provide empirical evidence demonstrating that the performance improvements are attributable to the novel aspects of the architecture rather than merely to an increase in the number of parameters.

3. Insufficient Discussion on Failure Cases: The paper could improve by providing more detailed discussions on any limitations or failure cases observed during the experiments. Understanding where the model falls short can be crucial for practical applications and future improvements.

**Suitability:**

2

---

### Official Review · Reviewer_91Uh · 2024-05-23

**Rating:** 5
**Confidence:** 3

**Summary:**

The proposed model EGLCR-Stereo utilizes edge structure information and multi-scale matching similarity features for better disparity estimation. It uses a lightweight network to predict the initial disparity, and then develops a multi-scale similarity feature extraction module to capture the fusion similarity information of stereo images across various scales. Moreover, it introduces an edge structureaware module to delineate edge information in complex scenes. Finally, it employs an iterative strategy for disparity estimation. Extensive experimental results demonstrate the effectiveness of the proposed model EGLCR-Stereo.

**Strengths:**

1. This work creatively enhances the model's ability to perceive depth edges by constraining the edge information generated from depth and semantics.
2. Extensive experimental comparisons and ablations are conducted on various widely-used stereo matching datasets, which demonstrate that the proposed framework EGLCR-Stereo consistently outperforms state-of-the art methods.

**Limitations:**

1. The proposed model EGLCR-Stereo should show the running distribution, e.g., model parameters, FLOPs and speed, etc.
2. In Table 1, the performance of RAFT lacks clear description.
3. Some bugs. In Formula 5, the definitions of some symbols are not given.

**Suitability:**

2

---

### Official Review · Reviewer_Vy1j · 2024-05-30

**Rating:** 4
**Confidence:** 4

**Summary:**

This paper proposes a method for disparity refinement to improve the performance of disparity maps by integrating edge, global, and local information. The method is validated on three datasets, ETH3D, Middlebury, and SceneFlow, and obtains pretty good results.

**Strengths:**

1. The method improves Raft-Stereo by extracting and fusing the edge and scale-adaptive information to make the features more discriminative. Therefore the matching results are better.
2. The method is validated on multiple datasets to demonstrate its generalization ability.

**Limitations:**

1. Didn't test the method on KITTI's online benchmark. It would be more convincing if this result were given.
2. The correlation between the edge guidance and the scale-adaptive attention is a bit poor, which makes this work feel a bit patchwork without a clear intention and backbone.

**Suitability:**

2

---

### Meta-Review · Area_Chair_jWVt · 2024-07-02

**Recommendation:** Accept (Poster)
**Confidence:** 4

**Metareview:**

The paper received mixed reviews. While it presents a promising approach to disparity estimation with clear strengths in robustness, generalization, and enhanced accuracy, it also has several limitations, including computational complexity, lack of novelty, and insufficient evaluation on additional benchmarks. Addressing these concerns, particularly through detailed discussions on computational costs, clearer differentiation from existing methods, and testing on the KITTI benchmark, will strengthen the paper.